# In Silico Searching for Alternative Lead Compounds to Treat Type 2 Diabetes through a QSAR and Molecular Dynamics Study

**DOI:** 10.3390/pharmaceutics14020232

**Published:** 2022-01-19

**Authors:** Nicolás Cabrera, Sebastián A. Cuesta, José R. Mora, Luis Calle, Edgar A. Márquez, Roland Kaunas, José Luis Paz

**Affiliations:** 1Department of Biomedical Engineering, Texas A&M University, College Station, TX 77843, USA; nicolascabrera93@tamu.edu (N.C.); rkaunas@tamu.edu (R.K.); 2Department of Chemistry, Manchester Institute of Biotechnology, The University of Manchester, 131 Princess Street, Manchester M1 7DN, UK; sebastian.cuesta@postgrad.manchester.ac.uk; 3Grupo de Química Computacional y Teórica (QCT-USFQ), Departamento de Ingeniería Química, Universidad San Francisco de Quito, Diego de Robles y vía Interoceánica, Quito 170901, Ecuador; 4Faculty of Pharmacy, University of Granada, 18011 Granada, Spain; lucho_calle_mendoza@hotmail.com; 5Facultad de Ciencias Médicas, Instituto de Investigación e Innovación en Salud Integral, Universidad Católica Santiago de Guayaquil, Guayaquil 09013493, Ecuador; 6Grupo de Investigaciones en Química y Biología, Departamento de Química y Biología, Facultad de Ciencias Exactas, Universidad del Norte, Carrera 51B, Km 5, vía Puerto Colombia, Barranquilla 081007, Colombia; 7Departamento Académico de Química Inorgánica, Facultad de Química e Ingeniería Química, Universidad Nacional Mayor de San Marcos, Cercado de Lima 15081, Peru; jpazr@unmsm.edu.pe

**Keywords:** free fatty acid receptor 1, type 2 diabetes, molecular dynamics, molecular docking, agonits of FFA1

## Abstract

Free fatty acid receptor 1 (FFA1) stimulates insulin secretion in pancreatic β-cells. An advantage of therapies that target FFA1 is their reduced risk of hypoglycemia relative to common type 2 diabetes treatments. In this work, quantitative structure–activity relationship (QSAR) approach was used to construct models to identify possible FFA1 agonists by applying four different machine-learning algorithms. The best model (M2) meets the Tropsha’s test requirements and has the statistics parameters R^2^ = 0.843, Q^2^_CV_ = 0.785, and Q^2^_ext_ = 0.855. Also, coverage of 100% of the test set based on the applicability domain analysis was obtained. Furthermore, a deep analysis based on the ADME predictions, molecular docking, and molecular dynamics simulations was performed. The lipophilicity and the residue interactions were used as relevant criteria for selecting a candidate from the screening of the DiaNat and DrugBank databases. Finally, the FDA-approved drugs bilastine, bromfenac, and fenofibric acid are suggested as potential and lead FFA1 agonists.

## 1. Introduction

Diabetes mellitus affected approximately 463 million people worldwide (~9.3% of the population) in 2019, with numbers expecting to rise to 578 million (~10%) by 2030 and 700 million (~11%) by 2045 [1]. Type 2 Diabetes Mellitus (T2DM) is the most common type of diabetes, representing 80% of all the cases [2]. T2DM is characterized by reduced secretion of insulin from β-cells and resulting hyperglycemia associated with cardiovascular complications such as cardiac ischemia and stroke [3,4]. Current methods to control blood sugar, including controlled diet, metformin, sulfonylurea, and insulin [5], have limited efficacy and are associated with potential health problems such as hypoglycemia, weight gain, and lack of sustained efficacy [6,7,8].

Free Fatty Acid Receptor 1 (FFA1), a protein expressed in pancreatic β-cell, has become a target of interest since FFA1 activation through ligand-receptor binding induces insulin secretion [7]. Unlike specific diet, metformin, and sulfonylurea, induction of insulin in response to FFA1 agonists is attenuated when blood glucose levels are excessive, providing negative feedback that reduces the risk of hypoglycemia [9].

The orthosteric drug TAK-875 is the most explored FFA1 agonist, showing a potent antidiabetic effect in early-stage clinical trials, with a lower propensity to cause hypoglycemia [10]. Unfortunately, the development of TAK-875 treatment was terminated in phase III clinical trials due to its hepatotoxicity [11]. Other FFA1 agonists have high lipophilicity, leading to poor pharmacokinetics, metabolic instability, toxicity, and off-target effects [12,13,14,15,16]. In 2008, Christiansen et al. discovered a series of 4-phenethynyldihydrocinnamic acid FFA1 agonists, the most potent being TUG-424; however, TUG-424 was too lipophilic to be a viable drug candidate [17]. This group subsequently synthesized several FFA1 agonists with relatively low lipophilicity, inspired by the molecular structure of TUG-424 and TAK-875 with varying levels of agonist activity [18,19,20]. While these studies provide important information, they lack a systematic analysis to relate agonist characteristics with their potency.

In silico analysis is an economic strategy for the exploration of new compounds [21,22]. Quantitative structure–activity relationship (QSAR) reveals relationships between structural descriptors and biological activities of chemical compounds. In this regard, a report of a multiple linear regression QSAR model to improve the FFA1 agonist activity was reported [23]. The biological activity was used as the logarithm of the half-maximal effective concentration (pEC_50_), and the best model was statistically well-validated in terms of leave-one-out cross validation. Furthermore, the X-ray crystallographic structure of the binding mode in the FFA1 was elucidated [24], and molecular dynamics simulation [8] and experimental studies [25] clarify the activation mechanism of FFA1 to treat diabetes. Strong hydrogen bond interactions with Tyr-91, Arg-183, Asn-244, and Arg-258 are linked with higher agonist potency on the activation of FFA1 [26]. Also, molecular dynamics were used to get insights in selectivity [27,28] and allosteric activation [29,30] to design new molecules to treat T2DM. These reports lay out valuable information to be used for in silico studies to find a possible FFA1 agonist.

In this article, new FFA1 agonists are proposed based on an in silico analysis. Lipophilicity and interaction with FFA1 of the suggested drugs were studied using QSAR, molecular docking, and molecular dynamics simulation. In this sense, a diverse dataset of 93 FFA1 agonists reported by Christiansen et al. [17,18,19,20] was used to build a model. The pIC_50_ of the molecules of the dataset, DiaNat [27], and DrugBank 5.1.7 (https://www.drugbank.ca/ (accessed on 18 February 2021)) [28] databases were predicted. In addition, a computational analysis of absorption, distribution, metabolism, and excretion (ADME) parameters was performed to explore lipophilicity, the effectiveness of the drug, and the affinity to reach the target site [29]. Finally, the crystal structure of FFA1 was used (PDB: 4PHU) for molecular docking and molecular dynamic simulations [30,31].

## 2. Materials and Methods

The step-by-step process of this study is shown in Figure 1.

### 2.1. Dataset

The molecular structure and experimental pEC_50_ values of 93 FFA1 agonists were collected from reports by Christiansen et al. [32]. The 3D structures of these molecules were further optimized based on Universal Force Field (UFF) molecular mechanics theory [32] with RDKit software (https://www.rdkit.org/ (accessed on 18 February 2021)) implemented in QuBiLs-MIDAS software v2.0. The simplified molecular-input line-entry system (SMILES) and pEC_50_ values of each compound are available in Appendix A. Topographic 3D molecular descriptors suggested for cheminformatics studies such as QSAR were calculated using QuBiLs-MIDAS [33]. A total of 3031 3D molecular descriptors were estimated based on many algebraic operations that consider linear, bilinear, and quadratic indexes for all molecules in the dataset [34]. Next, two subsets were obtained for the modelling process. The first subset with 1213 descriptors (SS_1213) was selected by applying Shannon entropy of 0.7 and Spearman coefficient of 0.7 as cutoffs. The Shannon entropy cutoff was used to identify descriptors to be removed when more than 30% of compounds have the same descriptor value, and the Spearman coefficient cutoff establishes the maximum allowable correlation between two descriptors. The second subset with 1393 descriptors (CSE_1393) was obtained by applying the attribute evaluator CfsSubsetEval implemented in Weka 3.8 [35], which evaluates the predictive ability of each descriptor in a supervised way for the response variable (i.e., pEC_50_). The compounds in the dataset were divided into training (75%) and test (25%) based on the Ward’s method of clustering to minimize the error of the sum of squares within clusters [36]; Ward’s method is a hierarchical cluster analysis for a rational split of the molecules into training and test set [37].

### 2.2. Variable Selection

Variable selection for the modelling process avoids overfitting, redundancy, and irrelevancy when models are constructed with few descriptors [38]. Variable selection was performed on Weka 3.8 with the wrapper method and the following machine learning techniques: multilinear regression (MLR), random forest, instance-based learning with *parameter k* (IBK), and Smola and Scholkopf’s algorithm for solving regression problem (SMOreg) [39]. The wrapper method uses a classifier to find a good set of descriptors by searching through the descriptor space and wraps a classifier in a cross-validation loop [40]. The aforementioned machine-learning techniques are more accurate than filter methods that evaluate the relevance of features based on high or low scores [41]. Then, the group of selected descriptors was assessed as possible models based on the statistical parameters obtained.

### 2.3. Applicability Domain

Applicability domain (AD) analysis is commonly performed to increase the confidence and reliability in predictions on QSAR models and is now considered a requirement for this type of modelling [42]. AD is the physicochemical, structural, biological space, or information on which the training set was developed and defines the space for reliable prediction of new drugs [43]. The AD in the current study was defined by the consensus strategy suggested in AMBIT discovery (http://ambit.sourceforge.net/ (accessed on 18 February 2021)). The AD methods used for the consensus are principal component analysis (PCA), range-based, probability density, Euclidean, and city block distance. The consensus score determines if a compound is inside or outside of the AD [44]. If three or more methods consider a compound as an outlier (score ≤ 0.25), that molecule is excluded from further dataset analysis.

### 2.4. Model Performance

Model performance was validated using well-known statistical parameters: coefficient of determination (R^2^), 5-fold cross-validation coefficient (Q^2^_CV_), the external validation coefficient (Q^2^_ext_), bootstrapping coefficient (Q^2^_boot_), Y-scrambling analysis, and Tropsha’s test [45] (available at www.oecd.org (accessed on 18 February 2021)). The functions of these statistical parameters are shown in Table 1. Also, the collinearity between the descriptors expressed as the Pearson’s correlation coefficient was evaluated.

### 2.5. Screening of the DrugBank 5.1.7 and DiaNat Databases

The QSAR model was employed to screen the DiaNat [27] and DrugBank 5.1.7 [28] databases in order to identify possible FFA1 agonists. DiaNat is composed of 336 natural products extracted from different medicinal plants. The antidiabetic activity of these natural products was tested either in vivo or in vitro. DrugBank 5.1.7 comprises 2636 approved drugs, 6127 experimental compounds, 118 nutraceutical compounds, and 245 withdrawn compounds. The candidates’ molecular structures were evaluated in detail and compared with dataset’s molecule structures.

### 2.6. Absorption, Distribution, Metabolism, and Excretion (ADME) Predictions

ADME parameters are routinely used to analyze the drug-likeliness and pharmacokinetics of potential drugs. In this study, SwissADME (http://www.swissadme.ch (accessed on 18 February 2021)) was used to predict ADME parameters for the identified FFA1 agonists. SwissADME canonicalizes, processes the SMILES of each molecule, adding hydrogens, neutralizing, and obtaining the Kekulé’s 3D structure.

Lipophilicity, the drug’s affinity for a lipid environment, is a key property in the design of possible FFA1 agonists, since high lipophilicity renders a drug candidate unsuitable [17]. Lipophilicity is typically expressed by the partition coefficient between *n*-octanol and water (log P_o/w_). A consensus log P_o/w_ was computed from the average values of five lipophilicity parameters: iLogP [46], XLogP3 [47], WLogP [48], MLogP [49], and SILICOS-IT [50]. The lipophilicity value obtained from an atomistic method on the fragmental system of Wildman and Crippen (WLOGP) and the topological polar surface area (TPSA) were used to construct the BOILED-Egg depictive model, which shows the drug probability of human intestinal absorption (HIA) and blood-brain barrier (BBB) permeation [51].

### 2.7. Molecular Docking

To gain insight into how the studied compounds fit into the target pocket and which amino acids are involved in the receptor–ligand interactions, a molecular docking analysis was performed targeting the orthosteric site of FFA1 based on its elucidated three-dimensional X-ray crystallographic structure (PDB:4PHU) [24]. The FFA1 structure was prepared by removing water molecules, ligands, and other external residues using Pymol [52]. AutoDockTools [53] was used to add polar hydrogens and rotatable bonds in the ligand. The centered coordinates for the grid box were obtained from the retrieved protein using Discovery Studio Visualizer V20.1.0 [54] (Xc = −50.057, Yc = −2.660, Zc = 57.856). Docking calculations were performed using AutoDock Vina using a grid box size of 24 × 18 × 18 (X, Y, Z, respectively), with default exhaustiveness, full ligand flexibility, and a 1 Å spacing [55].

### 2.8. Molecular Dynamics

Molecular dynamics simulations were performed on the most promising agonist candidates to gain insight into the protein–ligand interactions, including hydrogen bond formation and evaluating the stability of ligand-binding within the orthostatic site of FFA1. The conformations obtained from the docking calculation were used as the starting point to run the dynamic simulations using Gromacs 2019 [56]. First, the protein topology was built using AMBER99SB-ILDN [57] implemented in Gromacs 2019. Next, the ligands topology was built using antechamber python parser interface (ACPYPE) server and the Generalized Amber Force Field (GAFF) [58]. Finally, the system was solvated in a cube shape using the three-point water model (TIP3) and neutralized by adding sodium or chlorine atoms as required. Before the simulation, the system was relaxed and equilibrated. In the first equilibration, a constant number of particles, volume, and temperature (NVT) was chosen, and in the second equilibration, the number of particles, pressure, and temperature (NPT) was maintained constant. Equilibrations were done for 100 ps at 300 K using the Berendsen thermostat temperature coupling [59]. Then, the production of the trajectory of the molecular dynamic simulation was performed for 200 ns; the pressure coupling was set to 1 Barr and the thermostat coupling to 300 K [60].

### 2.9. Free Energy Calculations

The Molecular Mechanics Poisson–Boltzmann Surface Area (MM-PBSA) method was used to calculate the binding free energy of each candidate molecule to the FFA1 orthostatic binding pocket. The binding free energy was estimated using the g_mmpbsa tool [61] and Equation (1),
G_X_ = E_MM_ + G_polar_ + G_nonpolar_,(1)
where X is the ligand, receptor, or complex; E_MM_ is the vacuum molecular mechanics potential energy obtained for bonded and non-bonded interactions estimated using MM force field parameters; G_polar_ is the polar solvation energy calculated solving the Poisson–Boltzmann equation; G_nonpolar_ is the non-polar solvation energy obtained using the solvent-accessible surface area (SASA) model. These energies were obtained for the protein, ligand, and complex. The trajectory between 10 and 200 ns was used, taking snapshots every 5 ns.

## 3. Results and Discussion

### 3.1. Dataset and Variable Selection

Descriptors extracted from the 3D structures of the 93 compounds reported by Christiansen et al. [17,18,19,20] were used to generate predictive models of agonist activity; the descriptors of the best model were used to split the data into training and test sets. The pEC_50_ values cover more than 3 log activity distributions (from 4.79 to 8.04), which facilitates identification of descriptors that correlate with high agonist activity. The predictive models were obtained and evaluated with IBK, MLR, and Random Forest regression techniques based on the values of statistical parameters R^2^ and Q^2^_CV_ (see Appendix A). Models 1 and 2 (M1 and M2), obtained with MLR, have the highest R^2^ and Q^2^_CV_. Therefore, M1 (with 10 descriptors, Appendix A) and M2 (with 11 descriptors, Appendix A) were chosen for further analysis. A rational division of the molecules into training and test set was performed by applying the Ward’s method’s cluster analysis. The molecules were grouped into 10 clusters and approximately 25% were selected as a test set (Figure 1).

### 3.2. Applicability Domain

To detect outliers on the test set and provide reliable and accurate predictions, the applicability domain analysis was used. The applicability domain analysis used in this study is based on a consensus score of four methods. The score represents the fraction of the four methods that indicate a molecule is considered inside the applicability domain, with scores ranging from zero (molecule is identified as an outlier by all four methods) to one (molecule is inside the AD for all four methods) [21,62]. A score greater than 0.25 was used as the criteria for identifying compounds inside the AD. In M1, compounds 46 and 67 were identified as outside the AD and were removed from further analysis (Appendix A). In contrast, all molecules in M2 met the criteria (Appendix A).

### 3.3. Model Validation

The outliers were removed from the test set to evaluate the performance of M1 and M2. The performance of M1 and M2 was evaluated using R^2^, Q^2^_CV_, Q^2^_ext_, and mean absolute error (MAE) values. Values close to one for the statistic parameters R^2^, Q^2^_CV_, and Q^2^_ext_ indicate the models’ goodness-of-fit and predictability (Figure 2a). In addition, a reliable prediction is considered if the value of MAE is smaller than 0.1 times the training set range in logarithm units (MAE < 0.1 × (pEC_50-max_ − pEC_50-min_)). The MAE values of both models are smaller than 0.3, meeting the criteria for a reliable prediction [63] (Figure 2b).

The descriptors were calculated using physicochemical properties, which are largely used to understand the nature of the compound for drug development. The descriptors of M1 and M2 contain the following physicochemical properties: *A*log*P* (a), Charge (c), Electronegativity (e), Hardness (h), Polarizability (p), topological polar surface (psa), Refractivity (r), and Van der Waals volume (v). Figure 2c shows the number of times that property appears in M1 and M2 descriptors. *A*log*P*, present in both models, quantifies molecular lipophilicity, which is crucial for developing FFA1 agonists [17,18,19,20]. On the other hand, the frequencies for the physicochemical properties “e”, “p”, and “v” are substantially higher in M2 than in M1.

Additional statistical parameters demonstrate proper fitting of the models. Models M1 and M2 are presented in Equations (2) and (3), respectively. Q^2^_boot_ is the pEC_50_ prediction obtained by a based perturbation. The values of Q^2^_boot_ > 0.7 suggest that the dataset’s correlation fitting does not affect the perturbation. a(R^2^) and a(Q^2^) were obtained from Y-scrambling analysis and their small values suggest the pEC_50_ prediction does not occur by chance. M1 and M2 are well-fitted models based on the high value of the Fisher’s statistical test (F > 27), the slight standard deviation (s < 0.334), and the correlation between predicted and experimental pEC_50_ (Figure 3).
pEC_50-M1_ = 0.518 X_3D1_ − 0.132 X_3D2_ − 0.325 X_3D3_ + 0.002 X_3D4_ + 0.038 X_3D5_ − 0.024 X_3D6_ + 162.563 X_3D7_ + 7.409 X_3D8_ + 0.045 X_3D9_ + 0.222 X_3D10_ + 3.3864(2)
R^2^ = 0.872, Q^2^_boot_ = 0.792, F = 39.47, s = 0.298, a(R^2^) = 0.112, and a(Q^2^) = −0.269
pEC_50-M2_ = −0.858 Y_3D1_ + 0.056 Y_3D2_ + 0.090 Y_3D3_ + 31.113 Y_3D4_ + 0.175 Y_3D5_ − 1.512 Y_3D6_ − 0.009 Y_3D7_ + 0.620 Y_3D8_ − 4.26 Y_3D9_ − 0.069 Y_3D1O_ − 0.059 Y_3D11_ − 32.983(3)
R^2^ = 0.843, Q^2^_boot_ = 0.740, F = 27.74, s = 0.334, a(R^2^) = 0.130, and a(Q^2^) = −0.302

To analyze the collinearity of the descriptors, Pearson’s coefficients were calculated for pairwise comparisons of the descriptors in M1 and M2 and are available in Appendix A. The Pearson’s coefficients for M1 (from −0.501 to 0.408) and M2 (from −0.330 to 0.453) suggest non-collinearity between the models’ descriptors.

In addition, Tropsha’s test (available at www.oecd.org (accessed on 18 February 2021)) validates the accuracy of the prediction of pEC_50_ values for both models. M1 and M2 met all the test requirements, suggesting an accurate prediction (Table 2). The model that will be considered for further analysis is based on the coverage and predictability of the external test set. Therefore, based on the 100% coverage on the AD, M2 will be considered to predict the pEC_50_ for the screening databases.

### 3.4. Screening of the DrugBank 5.1.7 and DiaNat Databases

With the aim to find new drug candidates for treating T2DM, molecules from relevant databases DiaNat [27] and DrugBank 5.1.7 [28] were screened using M2. A total of 31 compounds from DiaNat-DB met our criteria for the applicability domain with pEC_50_ values estimated to be from 5.5 to 8.0. pEC_50_ values and the SMILES are available in Appendix A. The molecules with highest predicted affinity for FFA1 (pEC_50_ > 7.5) are nuciferine, γ-mangostin, morolic acid, curcumin, and 3-*O*-acetyloleanolic acid.

A total of 282 compounds from DrugBank 5.1.7 met our criteria for the applicability domain. Their predicted pEC_50_ values are from 3.77 to 9.35, and SMILES are available in Appendix A. It is important to note that 40 compounds of the DrugBank database have pEC_50_ values greater than 8, with the antipsychotic Haloperidol (DB00502), Vitamin K1 (DB01022), and carotenoid Zeaxanthin (DB11176) having the most significant values.

Further results about ADME predictions, molecular docking, molecular dynamics simulations of the most suitable FFA1 agonist candidates of the dataset (training/test set), and screening (DiaNat, and DrugBank) databases, are presented in the following sections.

### 3.5. Absorption, Distribution, Metabolism, and Excretion (ADME) Predictions

SwissADME (http://www.swissadme.ch (accessed on 18 February 2021)) was used to calculate physicochemical properties, lipophilicity, water-solubility, pharmacokinetics, drug-likeness, medicinal chemistry properties, and bioavailability score of the molecules. Bioavailability is a fast screening in the compounds to evaluate the possibility to be considered an oral drug [64]. All the compounds have a high bioavailability score of 0.85 in the dataset. The lipophilicity obtained from Wildman and Crippen method (WLOGP) and the topological polar surface area (TPSA) were used to construct the BOILED-Egg (Figure 4). The Boiled egg is a plot of WLOGP vs. TPSA and predicts the gastrointestinal absorption (HIA) and brain penetration (BBB). The four molecules (46, 50, 51, and 53) in the grey region are predicted to have poor intestinal absorption and brain penetration. P-glycoprotein substrates (Pg-p) is a restrictive barrier to maintain homeostasis of the brain and is key among ATP-binding cassette transporters. The red and blue dots are p-glycoprotein substrates (PGP+) and *P*-glycoprotein non-substrates (PGP−), respectively [65].

The ADME properties of the eight compounds in the dataset with the highest biological activity (pEC_50_ > 7.4), BBB permeation, and high HIA, are shown in Table 3 along with solubility and lipophilicity properties. The searching of possible FFA1 agonists with low lipophilicity has been highlighted by Christiansen et al., who focused on derivatives of TUG-424 (Molecule 15) [20]. Lipophilicity and water solubility have been shown to facilitate drug formulation and handling [66]. In this regard, Molecules considered for further analysis must have a log P_o/w_ value smaller than 3.83. Molecules 15, 52, 49, 48, and 47 are discarded for further investigation to be too lipophilic (Consensus Log P_o/w_ > 3.83). Water solubility was evaluated by three models, the first was ESOL [67]; the second was adapted from Ali et al. [68]; the third was developed by SILICOS-IT [66]. The qualitative scale to estimate water solubility is based on the log S scale: highly soluble > 0 > very soluble > −2 > soluble > −4 > moderately soluble > −6 > poorly soluble > −10 > insoluble. Compounds 91 (test set), 92 (training set), and 93 (training set) are either moderately soluble or soluble in at least two solubility methods. These compounds have smaller lipophilicity attributed to the incorporation of polar methoxy (compound 92) or cyanide (compounds 91 and 93) groups.

Regarding the compounds from the screening database, approximately 90.5% of the screened molecules are BBB permeable and have high HIA. Same as mentioned above for the dataset, the molecules in the screening to be considered for further analysis must meet high activity (pEC_50_ ≥ 7.4) and low lipophilicity (Consensus log P_o/w_ ≤ 3.83). Consequently, 25 compounds from the Drugbank and 1 compound from the DiaNat database met these requirements and were considered for further analysis.

### 3.6. Molecular Docking

Molecular docking was used to characterize the binding mode and steric fit of the ligands into the orthosteric site of FFA1, as well as to gain insights into protein–ligand interactions involved. Negative docking scores of the database compounds, ranging from −8.1 to −11.0 kcal/mol (Appendix A), suggest a favorable fit into the active site. The docking scores of the most potent compounds, i.e., 91 (−9.4 kcal/mol), 92 (−9.2 kcal/mol), and 93 (−10.3 kcal/mol), are comparable to the positive controls TAK-875 (−9.9 kcal/mol) and 15 (−10.2 kcal/mol).

The experimental configuration of TAK-875 bound to FFA1 (PDB: 4PHU) and the predicted docked configuration obtained in this work for the same compound were compared (Figure 5a). The molecular structure of the experimental structure is mostly superimposed with the docked configuration (RMSD = 1.90 Å), except at the methylsulfonyl tail of the molecule after the ether group that differs substantially. However, this is unlikely to be of consequence since the tail is very flexible and lies outside the orthosteric pocket, so it would not affect the activation occurring inside the active site. On the other end of the molecule, it is observed that the carboxylic acid of the docked structure presents a small shift compared to the experimental one.

The interactions of the positive controls occur mainly with Ala-83, Val-84, Tyr-91, Leu-138, Phe-142, Leu-171, and Arg-183. The interactions of TAK-875 and compound 15 with FFA1 are shown in Figure 5b and Appendix A, respectively. The carboxylic group in TAK-875 forms hydrogen bonds (HBs) with Tyr-91, Arg-183, and Arg-258. Arg183 and Arg-258 were reported in the literature as essential to anchor ligands containing carboxylate groups, while Tyr91 was reported to produce aromatic/hydrophobic interactions with the ligand [26].

Molecular docking was also performed on the 26 compounds resulting from the screening. The docking scores are presented in Appendix A. Seventeen compounds were taken out from the study because they stay outside the active site. Furthermore, suprofen and carbocromen were also removed due to their severe side effect leading to their market withdrawal. The docking scores of the remaining seven compounds (−8.5 to −10.3 kcal/mol) suggest favorable FFA1-ligand interactions (Appendix A). These compounds are anileridine (DB00913), a synthetic opioid; bromfenac (DB00963), a non-steroidal anti-inflammatory; bilastine (DB11591), an antihistamine drug; sulfinpyrazone (DB01138), an uricosuric; fenofibric acid (DB13873), a lipid-lowering agent conditions including hypertriglyceridemia, mixed dyslipidemia, and primary hyperlipidemia [69]; indacaterol (DB05039) used in situations such as asthma and chronic obstructive pulmonary diseases [70]; and curcumin (DBDB11672), an antioxidant, hypoglycemic, wound-healing, and antimicrobial compound [71].

The interactions between ligands and amino acids linked to FFA1 activation, high potency, and stabilization are shown in Figure 6. HBs are essential in the stabilization of the ligand–protein complex and FFA1 activation. Tyr-91, Arg-183, and Arg-258 were reported as relevant residues for the activation of the FFA1 [26]. Tyr-91 forms HB with compound 15, TAK-875, bilastine, and bromfenac. Arg-183 forms HBs with all compounds from the dataset and fenofibric acid. Arg-258 form HBs with compound 91, 92, TAK-875, and fenofibric acid. Interestingly, compounds 91, 92, and 93, the most actives compounds in the dataset, do not form hydrogen bonds with Tyr-91. On the other hand, Val-84 and Leu-138 seem to play an essential role during binding as they form π interactions with most of the compounds shown in Figure 6.

### 3.7. Molecular Dynamic Simulations

Molecular dynamics simulations were run for 200 ns to explore the trajectory of the binding process of five compounds from the dataset (15, 91, 92, 93, and TAK-875) and the seven compounds selected from the screening. To determine the reliability of the calculation along the trajectory, the changes in the potential energy, density, pressure, volume, temperature, and box size plots were analyzed.

The root-mean-square deviation (RMSD) was analyzed to explore the stability of the FFA1-ligand complex during the trajectory. Both the RMSD of the FFA1 (Figure 7) and the ligands (Figure 8) are expected to change during the simulation as the protein and the ligand reach a stable complex conformation. When the system reaches a stable configuration, fluctuations in the RMSD should decrease, meaning equilibrium is achieved. FFA1 reaches equilibrium in most of the FFA1-ligand complexes after 60 ns. The FFA1-TAK-875 complex was taken from experimental data and started from the equilibrium. The lowest RMSD values in the FFA1-TAK-875 complex validate the feasibility of the simulations.

The RMSD values of the ligands below the threshold (dotted line 0.30 nm) mean the optimized structures found on the docking analysis were appropriate (Figure 8). The RMSD of the ligands suggests the stability of the system during the trajectory. TAK-875 RMSD “jumps” between 50–100 ns and 185–190 ns due to a fluctuation on the methylsulfonyl tail. Curcumin RMSD goes above the threshold moving outside the active site during the simulation. Therefore, curcumin was taken out for further analysis.

Potent agonists showed interactions with amino acids that might be essential for the activation of FFA1. The number of hydrogen bonds (HBs) in the FFA1-ligand complex was analyzed along with the occupancy (number of times that a HB appears during the simulation) with residues Tyr-91, Arg-183, Asn-244, and Arg-258. Most of the compounds form between 1 to 6 HBs (Figure 9). Sulfinpyrazone comprises the highest number of HBs (24), and compound 92 forms the least. The occupancies confirm similar HB formation to the molecular docking study. Most of the compounds form interactions with Tyr-91, which is related to the activation of FFA1 and higher potency of the ligand [26]. However, compound 93, one of the most potent compounds tested in vitro, did not interact with Tyr-91. This compound has the highest occupancy with Arg-183 (63.0%). Asn-244 was reported to be important in anchoring the carboxylic acid [26]. Asn-244 did not appear in the docking study but presents important occupancies with compounds 15 (40%) and 93 (17%). Therefore, Arg-183 and Asn-244 may have an essential role in FFA1 activation.

Lennard-Jones energy and short-range Coulomb potential quantify the interaction energy of the protein–ligand complex (Appendix A). The short-range Lennard-Jones energy profile for all the models (−55 to −35 kcal/mol) indicates a similar interaction between the ligands and FFA1. Bilastine presented a −45 kcal/mol value during the first half and a value of −60 kcal/mol in the second half of the simulation, which suggests a more stable interaction was achieved in the second half. The ligands present a short-range Coulomb potential in accordance with the HBs formation during the dynamics. In this sense, compound 92 and indacaterol have the highest values, while compound 93 and bromfenac are the lowest. According to short-range Lennard-Jones energy and short-range Coulomb potential, all the compounds managed to interact favorably with FFA1, being possible agonist candidates.

The ligand–protein interaction was analyzed applying the MMPBSA approach [61], in which it is possible to estimate the free binding energy based on van der Waals, electrostatic, and solvent accessible surface area (SASA) energies (Table 4). The negative values found for the four energies agree that there is a favorable interaction between FFA1 and the ligands. Similar binding energy values were obtained for all the datasets’ systems, being TAK-875, the compound with the highest affinity (−30.88 kcal/mol). For the ligands of the screening, the range of the binding energy goes from −15 kcal/mol in bromfenac to −37 kcal/mol in bilastine. Although bromfenac presents the lowest binding energy value, its predicted pEC_50_ is the highest, comparing to TAK-875 (8.47 vs. 8.45).

The structures of the six compounds were compared to the TAK-875 structure to spot similarities and differences that could be essential for interaction at the active site (Figure 10). Interestingly, the carboxylic group in TAK-875 is also present in bromfenac, bilastine, and fenofibric acid. In anileridine, if a carboxyl group replaces the ester group, it may produce HBs, enhancing FFA1 activation. The TAK-875 benzene ring in the dihydrobenzofuran group interacts with Ala-83, Leu-138, and Leu-171 (Figure 5b). The benzene ring near the carboxyl group in bromfenac, bilastine, and fenofibric acid may produce the same interactions with those amino acids. Furthermore, anileridine, sulfinpyrazone, and indacaterol also present cyclic motifs in their structure that interact with Ala-83, Leu-138, and Leu-171. Finally, Val-84 that interacts with the second benzene ring in TAK-875 can interact with any cyclic structures on the left side of the six studies compounds.

Based on these results, bilastine presents the best characteristics, including the lowest binding energy (−36.97 kcal/mol), good lipophilicity (3.76), and a strong HB with Tyr-91 (71.9% occupancy) to be considered for in vitro analysis as an FFA1 agonist or as a lead compound to develop next-generation drugs against T2DM. Bromfenac is also an excellent option as a lead molecule due to its highest predicted pEC50 value. Bilastine and bromfenac are drugs of interest to treat diabetes-related diseases [72,73]. In vivo studies suggested bilastine as therapy for diabetic nephropathy, a common microvascular complication in patients diagnosed with diabetes mellitus [72]. Clinical trials reported the long-term benefits of bromfenac treating diabetic macular edema, a cause of loss of vision in patients with diabetes [73]. Finally, fenofibric acid is an interesting lead as changes may be introduced to produce tighter interactions with the FFA1 active site, developing a dual-target drug to treat T2DM and hypertriglyceridemia.

## 4. Conclusions

A dataset containing 93 compounds was split into training and test set and modelled employing different machine-learning techniques. M1 and M2 models, evaluated with MLR, showed the best statistical parameters: R^2^ = 0.872, Q^2^_CV_ = 0.812, and Q^2^_ext_ = 0.751, for M1; and R^2^ = 0.843, Q^2^_CV_ = 0.875, and Q^2^_ext_ = 0.850, for M2. Both models comply with the Tropsha’s test validation. M2 present the best coverage in the applicability domain analysis for the test set (100%). Therefore, it was employed for the screening of two relevant datasets (DiaNat and DrugBank). Based on a high activity (pEC_50_ ≥ 7.4) and low lipophilicity (Consensus log P_o/w_ ≤ 3.83) criterion, 26 compounds were selected as promising drugs. An exhaustive analysis on the protein–ligand interaction was done in the training, test set, and screened compounds using molecular docking and molecular dynamics tools. It was found that the interactions with Tyr-91, Leu-138, Leu-171, Arg-183, Ala-83, Val-84, Asn-244, and Arg-258 are crucial for FFA1 activation. The structure of the possible candidates presents different cores, which can be further explored to find new FFA1 agonists. These candidates are FDA-approved drugs, and future studies can explore their biological activity as FFA1 agonists.

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
