# Peer review of "In Silico Searching for Alternative Lead Compounds to Treat Type 2 Diabetes through a QSAR and Molecular Dynamics Study"

_pharmaceutics, 2022, doi:10.3390/pharmaceutics14020232_

Round 1

Reviewer 1 Report

Dear Authors,

Manuscript ID: pharmaceutics-1491706

In silico searching for alternative lead-compounds to treat type-2 diabetes. A QSAR and Molecular Dynamics study.

The work and results look good to way Authors presented.

Authors used quantitative structure-activity relationship (QSAR) approach 25 to construct models to identify possible FFA1 agonists by applying four different machine 26 learning algorithms.

New FFA1 agonists are proposed based on an in silico analysis.

M2 models evaluated with MLR showed the best statistical parameters: R2= 0.843, Q2CV= 0.875, and Q2ext= 0.850

Author well performed on exhaustive analysis on the protein-ligand interaction in the training, test set, and screened compounds using molecular docking and molecular dynamics tools.

Performed computational analysis of ADME parameters to explore lipophilicity, the effectiveness of the drug, and the affinity to reach the target site were well supported the proposed manuscript.

It is very interesting work. I would recommend publishing as is in the journal Pharmaceutics

Author Response

We really appreciate your positive comments

Reviewer 2 Report

The authors present an interesting but purely computational study on potential FFA1 binding ligands based on QSAR models and supported by Docking and MD simulations.

This study would be improved immensely if the computational data could be supported by experimental validation, however, since that is likely not feasible for the authors the theoretical validation should be more thorough. Also the background research and introduction need to be improved

  • Structural prerequisites of agonism and antagonism should be discussed and it should be made clear how the model is able to distinguish. How do known antagonists perform in the model?
  • The authors should describe the achieved RMSD value of the Re-docking of TAK-875
  • “The carboxylic group in 381 TAK-875 forms hydrogen bonds (HBs) with Tyr-91, Arg-183, and Arg-258. Arg183 and 382 Arg-258 were reported as essential to anchor ligand’s carboxylate groups, while Tyr91 383 was reported to produce aromatic/hydrophobic interactions with the ligand [27].”

There seems to be some Tyrosin confusion in this sentence. Is one of the two Tyrs Tyr 240 perchance?

  • A thorough literature review on the hit compounds needs to be conducted. Has any activity been reported on FFA1 or related proteins? How does the measured activity corresponds to the predicted activity.This would serve as a further step of validation for the author’s methods.
  • Which compounds are completely unknown on the target? Is there antidiabetic activity known?
  • “Interestingly, these candidates are FDA approved drugs and commercially available.”

It is not much of a surprise in the drugbank, which contains mainly approved drugs.

Author Response

The authors present an interesting but purely computational study on potential FFA1 binding ligands based on QSAR models and supported by Docking and MD simulations.

This study would be improved immensely if the computational data could be supported by experimental validation, however, since that is likely not feasible for the authors the theoretical validation should be more thorough. Also, the background research and introduction need to be improved

1.- Structural prerequisites of agonism and antagonism should be discussed and it should be made clear how the model is able to distinguish. How do known antagonists perform in the model?

Answer: The model presented in our work was built and evaluated for FFA1 agonists. FFA1 receptor is an agonist-bound R′ inactive-like state where molecules that interact with the active site must be agonists. The model is incapable of distinguishing structural differences between agonists or antagonists, and that issue is going to be analyzed in a further article.

2.- The authors should describe the achieved RMSD value of the Re-docking of TAK-875.

Answer: Thanks for the comment. The RMSD (1.90 Å) obtained was added to the manuscript and the main differences were discussed.

3.- “The carboxylic group in 381 TAK-875 forms hydrogen bonds (HBs) with Tyr-91, Arg-183, and Arg-258. Arg183 and 382 Arg-258 were reported as essential to anchor ligand’s carboxylate groups, while Tyr91 383 was reported to produce aromatic/hydrophobic interactions with the ligand [27].” There seems to be some Tyrosin confusion in this sentence. Is one of the two Tyrs Tyr 240 perchance?

Answer: Thanks for the comment. To clarify the interactions reported in the literature and obtained in our work the sentence was re-written. The first sentence refers to our findings while the second one is results found in the literature.

“The carboxylic group in TAK-875 forms hydrogen bonds (HBs) with Tyr-91, Arg-183, and Arg-258. Arg183 and Arg-258 were reported in the literature as essential to anchor ligand’s carboxylate groups, while Tyr91 was reported to produce aromatic/hydrophobic interactions with the ligand ” Line 382 to 385

4.- A thorough literature review on the hit compounds needs to be conducted. Has any activity been reported on FFA1 or related proteins? How does the measured activity correspond to the predicted activity? This would serve as a further step of validation for the author’s methods.

5.- Which compounds are completely unknown on the target? Is there antidiabetic activity known? “Interestingly, these candidates are FDA approved drugs and commercially available.” It is not much of a surprise in the drugbank, which contains mainly approved drugs.

Answer: Thank you for your comment. The sentence was re-written. “These candidates are FDA approved drugs and future studies can explore the biological activity as FFA1 agonists” (Added to the manuscript – Line 528 to 539). Pharmacodynamics and toxicity of the compounds from the screening have been reported. Therefore, the production of new antidiabetic drugs from these compounds will have a faster path towards reaching the market. 

Reviewer 3 Report

This study presents an in silico approach for identification of novel agonists against FFA1, a receptor associated with diabetes. A QSAR model is used to screen molecules from two well-known databases of compounds, validated accordingly, and further molecular dynamics and docking analyses are used to further fine tune molecular selection.

While there is not much novelty regarding methodology, the study is sound and all the methods are adequately presented. I have no major issues to report to the authors. I have only found some minor typographical issues that should be addressed:

L.112 25 missing %

L.123 parameter k should be in italics

L.168 same for n-octanol

Eq. 1, comma missing after the last term

L. 421 "The" incorrectly capitalized

Author Response

Thank you very much; we really appreciate your comments. All the paper was reviewed and all typos corrected